# Antioxidant Genetic Variants Modify Echocardiography Indices in Long COVID

**DOI:** 10.3390/ijms241210234

**Published:** 2023-06-16

**Authors:** Milika Asanin, Marko Ercegovac, Gordana Krljanac, Tatjana Djukic, Vesna Coric, Djurdja Jerotic, Marija Pljesa-Ercegovac, Marija Matic, Ivana Milosevic, Mihajlo Viduljevic, Goran Stevanovic, Jovan Ranin, Tatjana Simic, Zoran Bukumiric, Ana Savic-Radojevic

**Affiliations:** 1Faculty of Medicine, University of Belgrade, 11000 Belgrade, Serbia; masanin@gmail.com (M.A.); ercegovacmarko@gmail.com (M.E.); gkrljanac@gmail.com (G.K.); tatjana.djukic@med.bg.ac.rs (T.D.); vesna.coric@med.bg.ac.rs (V.C.); djurdja.jovanovic@med.bg.ac.rs (D.J.); marija.pljesa-ercegovac@med.bg.ac.rs (M.P.-E.); marija.matic@med.bg.ac.rs (M.M.); ivana.milosevic00@gmail.com (I.M.); goran_drste@yahoo.com (G.S.); njranin@gmail.com (J.R.); tatjana.simic@med.bg.ac.rs (T.S.); 2Clinic of Cardiology, Clinical Centre of Serbia, 11000 Belgrade, Serbia; mihajloviduljevic@gmail.com; 3Clinic of Neurology, Clinical Centre of Serbia, 11000 Belgrade, Serbia; 4Institute of Medical and Clinical Biochemistry, 11000 Belgrade, Serbia; 5Clinic of Infectious and Tropical Diseases, Clinical Centre of Serbia, 11000 Belgrade, Serbia; 6Department of Medical Sciences, Serbian Academy of Sciences and Arts, 11000 Belgrade, Serbia; 7Institute of Medical Statistics and Informatics, 11000 Belgrade, Serbia

**Keywords:** long COVID-19, antioxidant genetic variants, cardiological manifestations, SOD2, *GPX1*, *GPX3*, *Nrf2*

## Abstract

Although disturbance of redox homeostasis might be responsible for COVID-19 cardiac complications, this molecular mechanism has not been addressed yet. We have proposed modifying the effects of antioxidant proteins polymorphisms (superoxide dismutase 2 (*SOD2*), glutathione peroxidase 1 (*GPX1*), glutathione peroxidase 3 (*GPX3*) and nuclear factor erythroid 2-related factor 2, (*Nrf2*)) in individual susceptibility towards the development of cardiac manifestations of long COVID-19. The presence of subclinical cardiac dysfunction was assessed via echocardiography and cardiac magnetic resonance imaging in 174 convalescent COVID-19 patients. SOD2, *GPX1*, *GPX3* and *Nrf2* polymorphisms were determined via the appropriate PCR methods. No significant association of the investigated polymorphisms with the risk of arrhythmia development was found. However, the carriers of variant *GPX1**T, *GPX3**C or *Nrf2**A alleles were more than twice less prone for dyspnea development in comparison with the carriers of the referent ones. These findings were even more potentiated in the carriers of any two variant alleles of these genes (OR = 0.273, and *p* = 0.016). The variant *GPX* alleles were significantly associated with left atrial and right ventricular echocardiographic parameters, specifically LAVI, RFAC and RV-EF (*p* = 0.025, *p* = 0.009, and *p* = 0.007, respectively). Based on the relation between the variant *SOD2**T allele and higher levels of LV echocardiographic parameters, EDD, LVMI and GLS, as well as troponin T (*p* = 0.038), it can be proposed that recovered COVID-19 patients, who are the carriers of this genetic variant, might have subtle left ventricular systolic dysfunction. No significant association between the investigated polymorphisms and cardiac disfunction was observed when cardiac magnetic resonance imaging was performed. Our results on the association between antioxidant genetic variants and long COVID cardiological manifestations highlight the involvement of genetic propensity in both acute and long COVID clinical manifestations.

## 1. Introduction

Coronavirus disease 2019 (COVID-19) exhibits many manifestations of a systemic disease with important implications for the cardiovascular system [1]. Numerous findings support the relation between infection with severe acute respiratory syndrome coronavirus 2 (SARS-CoV-2) and increased prevalence of acute cardiovascular complications, including arrhythmia, heart failure (HF), ischemic heart disease, myocarditis and pericarditis. In addition, in long-term follow-up increased incidence of right ventricular dysfunction, myocardial fibrosis and hypertension was also determined. As one of the most frequent COVID-19 cardiac disorders found in up to 10.4% of patients with moderate to severe disease, arrhythmia was mostly represented as atrial fibrillation, non-sustained ventricular tachycardia and bradyarrhythmias [2,3,4]. It is important to note that recent data also suggested a relatively high incidence of hypertension in COVID-19 patients proposing an important role of endothelial dysfunction [5]. Moreover, almost 24% of patients who died from COVID-19 had characteristic symptoms of HF, potentiated similarities in the risk-associated profiles of patients with HF and severe form of disease [6,7,8]. However, to what extent these sequels are result of specific pathophysiological processes rather than direct consequences of acute infectious complications remains to be elucidated.

Multiple factors likely contribute to cardiac injury and dysfunction in COVID-19 [9]. It seems that the occurrence of COVID-19-induced myocardial damage is a multifaceted pathophysiological process that includes interplay between oxidative distress, inflammation, endothelial dysfunction and thrombosis as the major underlying mechanisms [10,11,12]. In this context, mitochondrial dysfunction along with excessive production of reactive oxygen species (ROS) has been pointed as a central hub responsible for disturbances in cellular homeostasis, metabolism and innate immune response. Interestingly, it seems that NLRP3 inflammasome presents as one of the critical players in these complex processes with consequent cardiac injury in COVID-19 [13]. Particularly, activation of this innate immune complex is necessary for the conversion of the cytokine precursors, pro-IL-1β and pro-IL-18, into their mature forms. This process results in the subsequent inflammatory cascade in SARS-CoV-2-induced inflammation of both myocytes and endothelial cells [14]. It further leads to myocarditis, loss of contractile function, altered ejection fraction, damage to cardiomyocytes and release of cardiac injury markers [15]. Additionally, any damage to the endothelium directly results in altered vascular tone and vasoconstriction, leading to ischemia, vascular inflammation, coagulation and thrombosis [16]. Thus, myocardial damage might constitute a risk factor for developing right ventricular (RV) dysfunction, as well as HF [17].

Based on the frequent presence of residual symptoms and organ dysfunction in post-acute COVID-19 patients, additional attention is now focused on long-term cardiovascular complications. Specifically, post-acute COVID syndrome (PACS) or the more commonly used “long COVID” represents considerable cause of subacute to chronic morbidity with a range of multiorgan symptoms which arise or persist beyond 4–12 weeks post-infection [18]. Moreover, interindividual differences observed in COVID-19-induced myocardial damage emerged recently as an important aspect to be clarified. The effect of the antioxidant genetic variations in the susceptibility and severity of acute COVID-19 was recently recognized. Among them, polymorphism rs6721961 of nuclear factor erythroid 2-related factor 2 (Nrf2) is found to reduce its transcription activity [19], whereas the presence of a variant superoxide dismutase *SOD2**Val rs4880 allele reduces the transport efficiency of newly synthesized enzyme in mitochondria by 30–40%, decreasing ROS scavenging in mitochondria [20,21]. Regarding polymorphisms of the glutathione peroxidases, *GPX1* rs1050450 and *GPX3* rs8177412, the presence of respected variant alleles results in lower antioxidant activity [22,23]. Furthermore, the influence of *SOD2* and *GPX1* polymorphisms on the inflammation and coagulation parameters in COVID-19 patients was also found [24].

Growing evidence clearly demonstrated cardiac dysfunction in the post COVID-19 patients, with special attention on an echocardiographic assessment that also provided several potential mechanisms of longer-term cardiovascular effects [25]. Since there is limited findings on the suggested role of disturbed redox homeostasis in the cardiac manifestation of long COVID, we aimed to investigate whether variations in antioxidant genetic profile, especially in genes encoding regulatory transcription factor Nrf2, as well as immediate (superoxide dismutase) and first-line defense (glutathione peroxidases) enzymes might be associated with cardiologic sequels in long COVID. We also put special emphasis on the potential association of these polymorphisms with the presence of subclinical cardiac dysfunction assessed via echocardiography in recovered COVID-19 patients.

## 2. Results

The clinical characteristics of 174 convalescent COVID-19 patients included in this study are summarized in Table 1. This cohort comprised 103 male patients (59.2%). Almost half of the recruited patients were normotensive (45.5%) and non-smokers (48.9%). Obese patients comprised 34.5%, whereas diabetes mellitus was present in 15.5% of the cohort. In terms of COVID-19 management, the majority of patients required hospitalization (82.2%). However, O_2_ support was indicated in only 36.8%. The distribution of assessed polymorphisms, occurring in genes encoding regulatory and key antioxidant proteins, is shown in Table 1. In addition, some of the basic cardiac findings and biomarker levels are also listed in Table 1.

The main cardiac indices concerning both left (end-diastolic left ventricle volume, end-diastolic left ventricle volume index, end-systolic left ventricle volume, systolic left ventricle volume index, end-diastolic left ventricle diameter, end-systolic left ventricle diameter, left ventricle ejection fraction, left ventricular mass, left ventricular mass index, and global longitudinal strain) and right ventricles (right fractional area change, right ventricle global longitudinal strain, and right ventricle ejection fraction), as well as left atrium (left atrium volume, and left atrium volume index) are shown in Table 2.

The risk of dyspnea or arrhythmia development was assessed via multivariate analysis with respect to *SOD2*, *GPX1*, *GPX3* and *Nrf2* genotypes (Figure 1 and Figure 2, respectively). The tree-plot in Figure 1 indicated no significant association of individual *SOD2*CT/TT*, *GPX1**CT/TT, *GPX3**TC/CC and *Nrf2**CA/AA genotypes with the risk of arrhythmia development (*p* > 0.05). In terms of dyspnea risk development, the carriers of *SOD2*CT/TT* exhibited 1.28 increased risk (95%CI:0.56–2.95, *p* = 0.56). On the other hand, the carriers of variant *GPX1**CT/TT, *GPX3**TC/CC and *Nrf2**CA/AA genotypes were more than twice less prone for dyspnea development in comparison with the carriers of referent *GPX1**CC, *GPX3**TT and *Nrf2**CC genotypes (Figure 2, *p* > 0.05). However, when the cumulative effect for the genotypes was computed, the carriers of any two genotypes (either *GPX1**CT/TT and *GPX3**TC/CC, or *GPX1**CT/TT, and *Nrf2**CA/AA, or *GPX3**TC/CC and *Nrf2**CA/AA) were at significantly lower risk for dyspnea development compared to referent genotypes (*GPX1**CC, *GPX3**TT and *Nrf2**CC; OR = 0.273, 95%CI:0.095–0.783, and *p* = 0.016).

The values of the measurement indicating the left ventricular end-diastolic diameter (EDD) and end-systolic diameter (ESD), end-diastolic volume (EDV), end-systolic volume (ESV), as well as the values for left ventricular mass index (LVMI) and longitudinal strain (SL peak G) were analyzed with respect to antioxidant genetic variants (Figure 3) and Nrf2. The carriers of *SOD2**CT/TT had significantly higher values of EDD (5.03 ± 0.55 vs. 4.80 ± 0.51, and *p* = 0.024), LVMI (87.14 ± 20.84 vs. 76.90 ± 15.91, and *p* = 0.005) and SL peak G (−19.15 ± 2.62 vs. −20.13 ± 2.37, and *p* = 0.036) when compared to the carriers of the referent *SOD2**CC genotype (Figure 3). On the other hand, the values of EDV were significantly lower in the carriers of *GPX3**TC/CC genotype compared to the carriers of *GPX3**TT genotype (101.13 ± 29.86 vs. 111.28 ± 29.27, and *p* = 0.05). As for *Nrf2* polymorphism, the carriers of either *Nrf2**CA or *Nrf2**AA genotype had increased EDV in comparison with the carriers of *Nrf2**CC genotype (116.02 ± 31.91 vs. 105.99 ± 28.36, and *p* = 0.057).

Regarding the right ventricle, the values of EF, GS and fractional area change (FAC) were analyzed with respect to *SOD2*, *GPX1*, *GPX3* and *Nrf2* genotypes (Figure 4). The carriers of *GPX3**TC/CC genotype had significantly higher values of EF in comparison to the carriers of *GPX3**TT reference genotype (64.12 ± 9.24 vs. 59.16 ± 10.63, and *p* = 0.007). Similarly, the values of FAC were significantly higher in the carriers of *GPX3**TC/CC genotype compared to the carriers of *GPX3**TT genotype (48.79 ± 8.82 vs. 44.59 ± 9.19, and *p* = 0.009). Finally, as indicated in Figure 5, the values of left atrial volume index (LAVI) were enlarged in the carriers of *GPX1**CT/TT genotype compared to the carriers of *GPX1**CC genotype (26.48 ± 6.53 vs. 24.27 ± 6.79, and *p* = 0.025).

No significant association between the investigated polymorphisms and cardiac disfunction was observed when cardiac magnetic resonance imaging was used.

In addition, the values of cardiac biomarkers (high-sensitive cardiac troponin T, hs-cTnT; brain natriuretic peptide, BNP; and D dimer) were analyzed with respect to the assessed genotypes, occurring in both regulatory and catalytic antioxidant proteins. Interestingly, the carriers of *SOD2*CT/TT* genotype had significantly higher values (*p* = 0.038) of hs-cTnT (median 7 (3–130) ng/L) compared to the carriers of *SOD2*CC* genotype (median 5 (3–17) ng/L). In addition, the difference in hs-cTnT values almost reached the level of statistical significance (*p* = 0.057) with the carriers of *GPX3**TT genotype having higher values of hs-cTnT (median 7 (3–130) ng/L) in comparison with the carriers of *GPX3**TC/CC (5 median (3–22) ng/L). When the levels of other cardiac biomarkers were assessed, no statistical significance was observed among carriers of different *SOD2*, *GPX1*, *GPX3* and *Nrf2* genotypes (*p* > 0.05).

## 3. Discussion

Based on the significant involvement of dysregulation of redox homeostasis in cardiac disorders, we have proposed modifying the effect of polymorphisms in catalytic and regulatory antioxidant proteins (*SOD2* rs4880, *GPX1* rs1050450, *GPX3* rs8177412, and *Nrf2* rs6721961) in individual susceptibility towards the development of cardiac manifestations of long COVID-19 even after mild to moderate forms of the disease. The data obtained herein have shown no significant association of individual *SOD2*, *GPX1*, *GPX3* and *Nrf2* polymorphisms with the risk of arrhythmia development. On the other hand, the carriers of variant *GPX1**T, *GPX3**C or *Nrf2**A alleles were more than twice less prone for dyspnea development in comparison with the carriers of the referent ones. These findings were even more potentiated in the carriers of any two variant alleles of these genes. Moreover, the variant *GPX* alleles were significantly associated with left atrial and right ventricular echocardiographic parameters, specifically LAVI, RFAC and RV-EF. Based on the relation between the variant *SOD2**T allele and the higher levels of LV echocardiographic parameters, EDD, LVMI and GLS, as well as troponin T, it can be proposed that recovered COVID-19 patients, who are carriers of this genetic variant, might have subtle left ventricular systolic dysfunction.

It has been well-documented that COVID-19 patients with pre-existing cardiovascular disorders were at a higher risk to develop severe cardiac injury [26]. In addition to worsening of ischemic heart disease and heart failure, COVID-19-induced myocardial injury with a range of diverse manifestations, including arrhythmia, acute coronary syndrome, myocarditis or cardiogenic shock has been found [27]. In our cohort of convalescent COVID-19 patients, dyspnea and arrhythmia were the most frequent cardiac long COVID-19 sequels. Indeed, based on our and previous studies, the presence of persistent dyspnea in more than a third of the recovered COVID-19 patients clearly imply subtle long-term cardiac changes. The extent of myocardial damage at mid- and long-term COVID-19 follow-up clinically manifested as persistent dyspnea could be consequence of several abnormalities, comprising mostly myocarditis-like patterns or ischemic injury [25]. It is now becoming clear that these post-COVID-19 cardiac manifestations are frequently accompanied by diastolic impairment, pulmonary hypertension, RV dysfunction, decrease in LV- and RV-longitudinal strain rather than systolic dysfunction [17]. Nevertheless, although major abnormalities of the LV function are not common [28], subtle cardiac changes attributed to SARS-CoV-2 infection cannot be entirely dismissed. In this setting, echocardiographic assessment of post-acute COVID-19 patients might contribute not only to the early detection of subclinical cardiac abnormalities, but also aid in discovery of potential mechanisms of longer-term cardiovascular effects.

There are several proposed mechanisms by which polymorphism of glutathione peroxidases, GPX1 and GPX3, might affect cardiac injury in long COVID 19. As the first line of enzymatic antioxidant defense system responsible for neutralizing hydrogen peroxide, GPX significantly contributes to preserving endothelial function and nitric oxide (NO) bioavailability [19,21,29]. Moreover, this protective activity is also important in prevention of oxidative posttranslational modifications of fibrinogen that could increase its thrombogenicity [30,31]. Experimental data pointed to the protective effect of overexpressed GPX1 towards accelerated thrombosis, by decreasing hydrogen peroxide-induced platelet hyperresponsiveness [32]. These findings are in line with those on the association of higher erythrocyte GPX1 activity with lower risk of cardiovascular events [33]. The previous findings on higher levels of both fibrinogen and D-dimmer in COVID-19 patients carriers of the variant *GPX1**T allele also implicate the consequential effect of this polymorphism on coagulation [24]. Beyond its antioxidant function, GPX1 might affect signaling cascades activated by SARS-CoV-2, specifically activation of ASK1-mitogen-activated protein kinase (MAPK), further interfering with the formation of the active ASK1 complex [34]. This way, it seems that GPX1 and GPX3 might also participate in the development of long COVID sequelae. Although results obtained in our study have shown that presence of at least one variant, *GPX1**T or *GPX3**C allele, reduces the odds of developing cardiac manifestations in longCOVID-19, such results are in concordance with the significant association of those genetic variants with the changes in the echocardiographic parameters of the right ventricle and the left atrium. These findings have additional importance, since the right ventricular dysfunction and the left atrial enlargement represent one of the most important features of left ventricular diastolic dysfunction [35]. Hence, it could be speculated that recovered COVID-19 patients, who are carriers of referent *GPX1* or *GPX3* alleles, are more prone to developing LV diastolic impairment.

Similarly, *SOD2* polymorphism had no effect on the susceptibility to cardiac COVID-19 sequels, dyspnea and arrhythmia. However, we found significant association between *SOD2**T allele and higher values of echocardiographic parameters of left ventricle, such as end-diastolic diameter, left ventricular mass index and LV longitudinal strain. These results further confirmed that recovered COVID-19 patients, who are carriers of the variant *SOD2* allele, probably showed subtle left ventricular systolic dysfunction based primarily on using the increased LV longitudinal strain as a measure of systolic function [36]. It seems that LV damage may be specifically attributed to long COVID-19 patients with no previous history of cardiac disease, but with significant increases in blood troponin, Tc, during the initial COVID-19 hospitalization [25]. Our results on significant association between the presence of variant *SOD2* allele and higher troponin T concentrations are in line with previous findings. It is important to note that lower LV global longitudinal strain is characteristic for majority of post COVID-19 patients with dyspnea when compared to asymptomatic patients, whereas LV ejection fraction is not significantly different. Although we did not find the modifying effect of *SOD2* polymorphism on the risk for dyspnea development, the findings of the present study emphasize the potential incremental value of *SOD2* genotyping to detect subclinical cardiac dysfunction in COVID-19 pateints. The *SOD2**T allele reduces the level of SOD2 enzyme in mitochondria, resulting in lower dismutation of superoxide anion into hydrogen peroxide and probably contribute to mitochondrial dysfunction [37,38]. Moreover, the accumulation of superoxide anion might also comprise a proinflammatory effect [39]. These differences was also observed in higher levels of proinflammatory cytokines, IL-1, IL-6, TNF-a, and IFN-γ, determined in peripheral blood mononuclear cells from the carriers of variant *SOD2**Val/Val genotype compared to those with referent *SOD2**allele [40]. Our data align with these findings, since we could hypothesize that in COVID-19 patients, carriers of *SOD2**T genetic variant during SARS-CoV-2 infection with larger inflammatory damage to cardiomyocytes was present.

Apart from antioxidant enzymes, we suggested that polymorphism of main antioxidant transcription factor Nrf2 also might have a certain impact on cardiac manifestations in long COVID-19 [20]. In this study, we found only subtle changes in several echocardiographic parameters regarding *Nrf2* polymorphism. This is not surprising keeping in mind that the majority of findings suggested that, during SARS-CoV-2 infection, silencing of Nrf2 signaling pathway is interconnected with consequent NF-kB signaling activation [41]. In addition to its regulatory role in synchronized induction of genes encoding antioxidant enzymes [42], this transcription factor may also exhibit anti-inflammatory effects, especially regarding the down-regulation of numerous proinflammatory cytokines [41].

Certain limitations could be considered in our study. This study’s findings may be influenced by potential biases arising from the relatively small number of participants and polymorphisms studied. Future study designs could benefit from increasing the number of participants, which would ultimately facilitate a more thorough analysis in terms of specific subgroups (e.g., age-depended) as well as by introducing the case–control design. Unfortunately, the obtained results cannot be translated to the other races as the study only included Caucasian people.

Our results on the association between antioxidant genetic variants and cardiological manifestations in long COVID highlight the involvement of genetic propensity in both acute and long COVID clinical manifestations and might contribute to better identification of potential therapeutic strategies for both the prevention and treatment of long COVID cardiac sequelae.

## 4. Materials and Methods

### 4.1. Patients and Study Design

This is a prospective observational cohort study of 174 COVID-19 patients (103 men and 71 women, with an average age of 55.65 ± 12.33 years) treated in the University Clinical Centre of Serbia, between July 2020 and February 2021. All participants were Caucasians by ethnicity. Inclusion criteria were (i) willingness to participate and provide informed consent, (ii) the absence of any prior cardiac disease history, and (iii) positive SARS-CoV2 reverse transcription (RT)-PCR test performed from nasopharyngeal and oropharyngeal swabs according to World Health Organization guidelines and using available RT-PCR protocols, age ≥ 18 years old. Exclusion criteria were (i) unwillingness to participate, and (ii) patients with ineligible echocardiography exams. The principles of the International Conference on Harmonization (ICH) Good Clinical Practice, the “Declaration of Helsinki”, and national and international ethical guidelines were followed during this study with the approval obtained from the Ethics Committee of the University Clinical Centre of Serbia. Clinical, demographic and epidemiological data were collected using the RedCap^®^ questionnaire (RedCap^®^. Available online: https://redcap.med.bg.ac.rs/, accessed on 19 May 2023, Faculty of Medicine, Belgrade, Serbia, used for project AntioxIdentification).

The data on medical history, signs and symptoms of the disease, comorbidities and laboratory parameters were obtained from the patients’ clinical records. Follow-up cardiological examination for all participants were considered eligible after a minimum of 1 month from the original diagnosis if they had resolution of respiratory symptoms and negative results on a swab test at the end of the isolation period. Meantime of follow-up cardiological examination were 6.1 ± 2.7 months after acute phase of COVID-19. All patients underwent on-site clinical examination, laboratory analysis and echocardiography.

### 4.2. Echocardiographic Examination

All resting standard echocardiographic examinations were performed using Vivid E95 (General Electric, Boston, MA, USA). Data were acquired via a 3.5 MHz transducer in the parasternal (long- and short-axis views) and apical views (two- and four-chamber and apical long-axis views). Echocardiographic methods were M-mode, 2D, color Doppler, pulse Doppler, continue Doppler, tissue Doppler and speckle-tracking imaging. All definitions and rules for measurements were in accordance with the recommendations of the European and American Society of Echocardiography [43].

The left ventricle (LV) volumes were measured from the apical two- and four-chamber views, and LV ejection fraction (EF) was calculated using Simpson’s rule [43]. LV mass index was calculated using the adequate formula [43]. LV diastolic function was evaluated using the recommendation of the American Society of Echocardiography (ASA) and the European Association of Cardiovascular Imaging (EACVI) [44,45].

Right ventricle (RV) and left atrium (LA) assessment was in accordance the ASA and EACVI [44,45,46].

The two-dimensional speckle tracking echocardiography (2D-STE), as a non-invasive ultrasound imaging technique was used for an objective and quantitative evaluation of global and regional myocardial deformation. Moreover, 2D-STE was used for the evaluation of systolic and diastolic myocardial function of LV, left atrium (LA) and right ventricle (RV). The recordings with a frame rate between 50 and 70 frames/s were performed and analyzed offline using General Electric software (EchoPAC software version 203 GE Medical Systems). All 2D-STE parameters of myocardial longitudinal strain were calculated offline in accordance with existing recommendations [43,44]. Longitudinal strain of LV is analyzed via the 18-segment segmentation model.

### 4.3. Cardiac Magnetic Resonance Data Acquisition and Postprocessing

Cardiac magnetic resonance (CMR) imaging was performed via the clinical 1.5-T scanner (Siemens Avanto, Siemens, Munich, Germany), using standardized and unified imaging protocols (University Clinical Center of Serbia, Center of CMR). CMR was conducted using standard protocol for morphological and functional assessment, late gadolinium enhancement (LGE), T1 and T2 mapping were conducted using MOLLI sequence, before and after contrast media application.

Steady-state free precession cine imaging, T2 mapping, pre- and post-contrast T1 mapping, T2-weighted Short-tau Triple Inversion Recovery (T2-STIR), first-pass perfusion, and late gadolinium enhancement (LGE) were acquired. Myocardial T1 and T2 mapping were acquired in a 4-ch, 2-ch and 3-ch long-axis directions as well as 3 short-axis slices (base level, midventricular and apex level) using a validated variant of a modified Look-Locker Imaging sequence (University Clinical Center of Serbia, Center of CMR, MOLLI). Myocardial T1 and T2 relaxation times were compare with the values of T1 and T2 relaxation times of the healthy control group as referent values of the CMR laboratory. Late gadolinium enhancement imaging was performed approximately 10 min after the administration of 0.1 mmol/kg of body weight of gadobutrol (Gadovist; Bayer, Leverkusen, Germany). Interpretation of LGE images followed the standardized postprocessing recommendations; myocardial LGE was visually defined by 2 observers based on the presence and predominant pattern as ischemic or nonischemic. Pericardial LGE was considered present when enhancement involved both pericardial layers, irrespective of the presence of pericardial effusion.

CMR results were extracted offline via Syngo software (Siemens Medical Solutions, https://www.siemens-healthineers.com/en-us, accessed on 18 May 2023), using a manual adjustment of the ventricular contours applied to determine LV mass, end-diastolic volumes, and ejection fractions.

Ventricular volumes and LV mass were indexed to body surface area, and the LV mass/end-diastolic volume ratio was used to assess LV concentric remodeling. Myocardial T1 and T2 mapping were determined as regions of interest (ROI) to find the highest values pre and postcontrast in the LV myocardium segments. The myocardial extracellular volume (ECV), expressed as % myocardium volume, was conventionally computed from (i) the T1 values from the pre-contrast MOLLI sequence (ii) the T1 values from post-contrast MOLLI sequence (acquisition scheme: 4(1)3(1)2) acquired 10–15 min after the injection and (iii) individual hematocrit values obtained from the blood sampled just before CMR. The formula of ECV (%) was 100% × (1-hematocrit) × [(1/postcontrast T1 myocardium) − (1/nativeT1myocardium)]/[(1/ postcontrast T1 blood) − (1/nativeT1 blood)] [47,48,49].

### 4.4. Genotyping

DNA isolation was performed on the EDTA-anticoagulated peripheral blood obtained from the study participants using PureLink™ Genomic DNA Mini Kit (ThermoFisher Scientific, Waltham, MA, USA). The assessed polymorphisms (*GPX1* rs1050450, *SOD2* rs4880, *GPX3* rs8177412, and *Nrf2* rs6721961) were determined as previously described [24].

Statistical data analysis was performed using IBM SPSS Statistics 22 (SPSS Inc., Chicago, IL, USA). Results were presented with regard to the data normality distribution, and appropriate statistical tests were chosen accordingly. Multivariate logistic regression was computed for calculating the odds ratio (OR) and 95% confidence interval (95%CI) in order to determine the potential association between the assessed genotypes and the odds for the particular cardiac outcome development. All *p*-values less than 0.05 were considered significant.

## Figures and Tables

**Figure 1 ijms-24-10234-f001:**
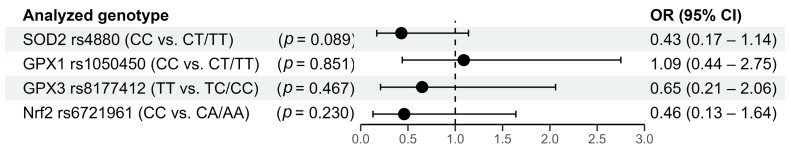
The association of antioxidant genetic variations and long COVID arrhythmia.

**Figure 2 ijms-24-10234-f002:**
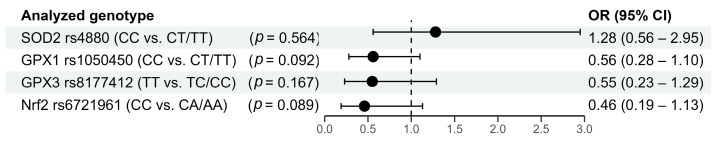
The association of antioxidant genetic variations and long COVID dyspnea.

**Figure 3 ijms-24-10234-f003:**
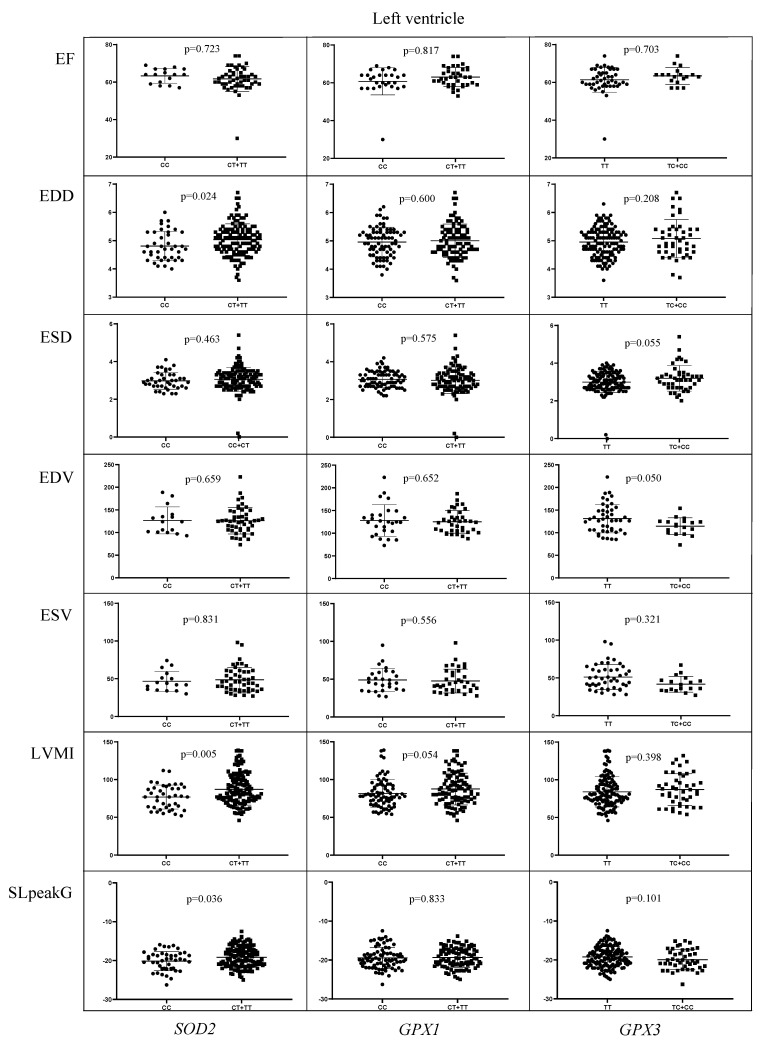
The values of the left ventricular ejection fraction (EF), the left ventricular end-diastolic diameter (EDD) and end-systolic diameter (ESD), end-diastolic volume (EDV), end-systolic volume (ESV), as well as the values for left ventricular mass index (LVMI) and longitudinal strain (SL peak G) with respect to *SOD2*, *GPX1* and *GPX3* genotypes.

**Figure 4 ijms-24-10234-f004:**
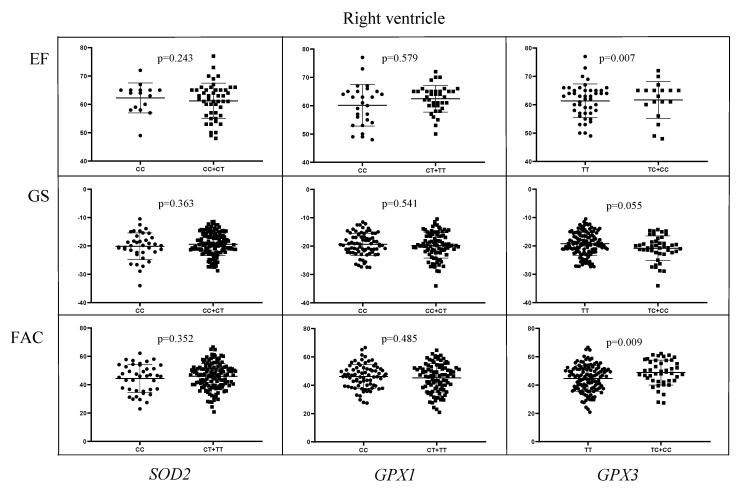
The values of right ventricle EF, GS and fractional area change (FAC) with respect to *SOD2*, *GPX1* and *GPX3* genotypes.

**Figure 5 ijms-24-10234-f005:**
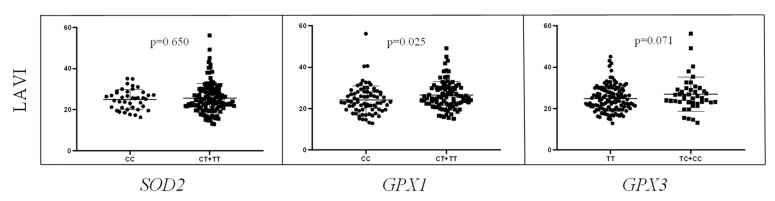
The values of left atrial volume index (LAVI) with respect to *SOD2*, *GPX1* and *GPX3* genotypes.

**Table 1 ijms-24-10234-t001:** Cardiac manifestations in 174 convalescent COVID-19 patients.

Overall Characteristics		Cardiac Characteristics	
Age (years) ^a^	55.65 ± 12.33		
Gender, n (%)		Arrhythmia, n (%) ^b^	
Male	103 (59.2)	No	153 (87.9)
Female	71 (40.8)	Yes	21 (12.1)
Hypertension, n (%) ^b^		Dyspnea, n (%) ^b^	
No	95 (54.5)	No	128 (73.6)
Yes	79 (45.5)	Yes	46 (26.4)
Obesity, n (%) ^b^		Blood pressure (mmHg) ^a^	
BMI < 30	114 (65.5)	Systolic	128.32 ± 19.47
BMI > 30	60 (34.5)	Diastolic	78.52 ± 11.15
Smoking, n (%) ^b^		hs-cTnT (ng/L) ^c^	7 (3–130)
Never	86 (48.9)		
Former	60 (34.1)		
Ever	17 (9.7)		
Diabetes, n (%) ^b^			
No	147 (84.5)	BNP (pg/mL) ^c^	18 (0–358)
Yes	27 (15.5)		
Hospitalization, n (%) ^b^			
No	31 (17.8)		
Yes	143 (82.2)		
Pneumonia, n (%) ^b^		D-dimer (mg/L) ^c^	0.34 (0.17–3.39)
No	24 (13.8)		
Yes	150 (86.2)		
O_2_ support, n (%) ^b^		CRP (mg/L) ^c^	1.55 (0.6–44.6)
No	110 (63.2)		
Yes	64 (36.8)		
Genotypes			
*SOD2 rs4880*, n (%) ^b^			
CC	40 (23.1)		
CT + TT	133 (76.9)		
*GPX1 rs1050450*, n (%) ^b^			
CC	77 (44.8)		
CT + TT	95 (55.2)		
*GPX3 rs8177412*, n (%) ^b^			
TT	128 (74.4)		
TC + CC	44 (25.6)		
*Nrf2 rs6721961*, n (%) ^b^			
CC	128 (74.9)		
CA + AA	43 (25.1)		

^a^ mean ± standard deviation, ^b^ percentage, ^c^ median (minimum-maximum); BMI—body mass index; CRP—C reactive protein; hs-cTnT—high-sensitive cardiac troponin T; BNP—brain natriuretic peptide; SOD—superoxide dismutase; GPX—glutathione peroxidase; Nrf2—Nuclear factor-erythroid factor 2-related factor 2.

**Table 2 ijms-24-10234-t002:** The main cardiac indices of 174 convalescent COVID-19 patients.

**Left Ventricle**	
End-diastolic volume (EDV, mL) ^a^	126.34 ± 28.93
End-diastolic volume index (EDVI, mL/m^2^) ^a^	64.15 ± 12.94
End-systolic volume (ESV, mL) ^a^	48.38 ± 15.45
End-systolic volume index (ESVI, mL/m^2^) ^a^	24.08 ± 6.57
EF (%) ^a^	61.98 ± 6.07
**Right ventricle**	
End-diastolic volume (EDV, mL) ^a^	125.30 ± 33.20
End-diastolic volume index (EDVI, mL/m^2^) ^a^	63.00 ± 14.47
End-systolic volume (ESV, mL) ^a^	49.03 ± 16.92
End-systolic volume index (ESVI, mL/m^2^) ^a^	25.05 ± 7.07
EF (%) ^a^	61.57 ± 6.03
**T1 mapping**	
Pre-contrast (ms) ^a^	1028.28 ± 108.58
Increased T1 pre-contrast values (%)	20.0
Postcontrast (ms) ^a^	452.59 ± 96.69
Increased T1 postcontrast values (%)	23.1
ECV (%) ^a^	22.9 ± 4.9
**T2 mapping**	
Pre-contrast (ms) ^a^	48.08 ± 7.00
Increased T2 pre-contrast values (%) ^b^	18.8
Postcontrast (ms) ^a^	54.59 ± 6.66
Increased T2 postcontrast values (%) ^b^	14.3
**LGE**	
Pathological findings (%) ^b^	44.4
**T2w**	
Pathological findings (%) ^b^	5.7
**Pericardial**	
Pathological findings (%) ^b^	12.7

^a^ mean ± standard deviation, ^b^ median (minimum-maximum).

## Data Availability

The data presented in this study are available on request from the corresponding author. The data are not publicly available due to privacy and ethical issues.

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
