# Peer review of "Antioxidant Genetic Variants Modify Echocardiography Indices in Long COVID"

_ijms, 2023, doi:10.3390/ijms241210234_

Round 1

Reviewer 1 Report

The subject of the study was the analysis of cardiovascular sequelae caused by long-term COVID-19. The work is part of the ongoing research on the long-term sequelae of this disease.
As a result, the influence of the Nrf2 antioxidant polymorphism on the occurrence of cardiac symptoms in the long-term course of COVID-19 was suggested. The contribution of genetic susceptibility to both acute and long-term clinical manifestations of COVID was also indicated. This knowledge may help better identify potential therapeutic strategies for preventing and treating cardiac sequelae caused by COVID-19.

The results presented in this study provide a good starting point for further research in this direction.

The publication is well written, and the results may interest many scientists, especially those working on similar research topics.

Author Response

Belgrade, 7th of June 2023

            Thank you for your interest in publishing our manuscript entitled “Antioxidant genetic variants modify echocardiography indices in long-COVID”. We have made all the changes in the manuscript as suggested by the reviewers. Thank you once again for your time and efforts. All corrections in the revised version of the manuscript are highlighted (track changes).

Yours sincerely,

Ana Savic-Radojevic and Zoran Bukumiric

ANSWERS TO REVIEWER NUMBER 1

We thank the reviewer for the time and efforts put to improve to quality of our manuscript.

Reviewer 2 Report

No suggestions

Author Response

Belgrade, 7th of June 2023

            Thank you for your interest in publishing our manuscript entitled “Antioxidant genetic variants modify echocardiography indices in long-COVID”. We have made all the changes in the manuscript as suggested by the reviewers. Thank you once again for your time and efforts. All corrections in the revised version of the manuscript are highlighted (track changes).

Yours sincerely,

Ana Savic-Radojevic and Zoran Bukumiric

ANSWERS TO REVIEWER NUMBER 2

We thank the reviewer for the time and efforts put to improve to quality of our manuscript.

Reviewer 3 Report

 Asanin et al investigated whether antioxidant genes are associated with cardiovascular abnormalities observed in long covid-19 syndrome. Long Covid genetics is complex and difficult to diagnose its symptoms.  The attempt for investigating antioxidant genes for cardiovascular abnormalities is commendable. Before accepting this manuscript for publication authors should answer and correct the following concerns.

1.       For each subtype, patient numbers are too less to obtain significance. Although it is a good attempt, authors should look for increasing patient numbers for future or follow up studies.

2.       In table 1, when genes are listed for genotypes, authors should mention the SNP ID along with gene name. Also, this portion could be separated from Table 1 to a separate table that would include p-value and odd ratio, CI etc.

3.       In figure 2 and 3, again the SNP ID is missing although mentioned in the text. But it is appropriate to mention in side by side of the gene, so will be easier to view for the reader. P-value also should be shown for each row with OR and CI.

4.       In Figure 3 and 4 legend, EF is not mentioned.

5.       As the author did not perform any PCA analysis, they should mention the race and other characteristics of the patients and controls.

6.       Whether authors obtained any effect of these SNPs in age dependent (55-62, 62-68) covid-19? Did they analyze it?

Quality of English is fine except some minor editing and grammer required.

Author Response

Belgrade, 7th of June 2023

            Thank you for your interest in publishing our manuscript entitled “Antioxidant genetic variants modify echocardiography indices in long-COVID”. We have made all the changes in the manuscript as suggested by the reviewers. Thank you once again for your time and efforts. All corrections in the revised version of the manuscript are highlighted (track changes).

Yours sincerely,

Ana Savic-Radojevic and Zoran Bukumiric

Please, find below a point-by-point answer to the reviewer’s comments:

ANSWERS TO REVIEWER NUMBER 3

Point 1. For each subtype, patient numbers are too less to obtain significance. Although it is a good attempt, authors should look for increasing patient numbers for future or follow up studies.

ANSWEWR 1: We thank the reviewer for the comment. We are aware that the number of observed units might stand in the way for obtaining the proper significance. However, we feel that this pilot attempt may be good in terms of postulating the aims of the future studies. Prospectively, we will amend this drawback by including more participants for each subtype. The aforementioned drawback was noted in the study Limitations by adding the following text

“Certain limitations could be considered in our study. The study findings may be influenced by potential biases arising from relatively small number of participants and polymorphisms studied. The future study design could benefit from increasing the number of participants, which would ultimately provide more thorough analysis in terms of specific subgroups (eg. age-depended) as well as by introducing the case-control design.” 

Point 2.1. In Table 1, when genes are listed for genotypes, authors should mention the SNP ID along with gene name.

ANSWEWR 2.1: We thank the reviewer for the careful observation. The SNP ID are now included in the Table 1 along with the gene name.

Point 2.2. Also, this portion could be separated from Table 1 to a separate table that would include p-value and odd ratio, CI etc.

ANSWER 2.2: Although this paper represents the continuation of the project study that tackled the role of polymorphisms encoding the antioxidant proteins in terms of assessing their role as risk biomarkers, this particular time we performed a prospective observational cohort study that included only 174 COVID-19 patients.  The future follow up studies could prospectively include larger number of participants, dichotomized into cases and respective controls. Therefore, a newly designed, case-control study can be performed to assess the role of the analyzed polymorphism not only between the affected patients subgroups, but compared to healthy matched controls. For the time being, it was impossible to recruit such control participants. The aforementioned drawback was assessed in the study limitation and the following text was added “The future study design could benefit from increasing the number of participants, which would ultimately provide more thorough analysis in terms of specific subgroups (eg. age-depended) as well as by introducing the case-control design”  

Point 3. In figure 2 and 3, again the SNP ID is missing although mentioned in the text. But it is appropriate to mention in side by side of the gene, so will be easier to view for the reader. P-value also should be shown for each row with OR and CI.

Answer 3. We thank the reviewer for the comment. In the designated figures, we added the SNP ID along with the gene name. Additionally, for each row, a P value was added.

Point 4. In Figure 3 and 4 legend, EF is not mentioned.

Answer 4. We thank the reviewer for the careful observation. The legends for figures 3 and 4 are corrected now.

Point 5. As the author did not perform any PCA analysis, they should mention the race and other characteristics of the patients and controls.

Answer 5. We thank the reviewer for the comment. All participants in our study were Caucasians by ethnicity. Therefore, there was no need to take this feature into account when the statistical analysis was performed. The characteristics of study participants were mentioned in the Manuscript, Methods section. However, this drawback was addressed in the Limitation section by adding the following text “Unfortunately, the obtained results cannot be translated on the other race as the study only included the white race

Point 6. Whether authors obtained any effect of these SNPs in age dependent (55-62, 62-68) covid-19? Did they analyze it?

Answer 6. We thank the reviewer for this notion. In this study the effect of SNPs in age dependent groups was not analyzed, however, we highly agree it would be interesting to see those results in our future studies with increased number of patients. The aforementioned drawback was assessed in the study limitation and the following text was added “The future study design could benefit from increasing the number of participants, which would ultimately provide more thorough analysis in terms of specific subgrupus (eg. age-depended) as well as by introducing the case-control design” 
